# Renal and hepatic function is preserved following inducible knockout of kynurenine pathway enzymes KMO or QPRT in adult mice

Benjamin S. Summers[1,2], Luke Milham[2,3], Sonia Bustamante[4], Krishan Gondal[1,2], Peggy Rentsch[2,3], Gayathri Sundaram[1,2¤], Michael D. Lovelace [1,2], Bryce Vissel[2,3], Bruce J. Brew [1,2,5,6]*

**1** Applied Neurosciences Program, Peter Duncan Neurosciences Research Unit, St. Vincent's Centre for Applied Medical Research, Sydney, New South Wales, Australia, **2** Faculty of Medicine and Health, School of Clinical Medicine, UNSW Sydney, Sydney, New South Wales, Australia, **3** Centre for Neuroscience and Regenerative Medicine, St. Vincent's Centre for Applied Medical Research, St Vincent's Hospital, Sydney, New South Wales, Australia, **4** Bioanalytical Mass Spectrometry Facility, Mark Wainwright Analytical Centre, UNSW Sydney, Sydney, New South Wales, Australia, **5** Departments of Neurology and Immunology, St. Vincent's Hospital, Sydney, New South Wales, Australia, **6** University of Notre Dame, Darlinghurst, Sydney, New South Wales, Australia

¤ Current address: Bionyeri Pty Ltd, Hornsby, Sydney, NSW, Australia
* b.brew@unsw.edu.au

## Abstract

The kynurenine pathway (KP) is the canonical route by which tryptophan is metabolised, almost all of which occurs in the liver, with significant expression of its enzymes also known in the kidney. We generated two novel mouse models for inducible global knockout of midpoint KP enzyme kynurenine-3-monooxygenase (KMO) and endpoint enzyme quinolinate phosphoribosyltransferase (QPRT; converts known neurotoxic KP metabolite Quinolinic acid to nicotinamide adenine dinucleotide (NAD) precursor via the de novo synthesis pathway). The KP is dysregulated in many renal and hepatic disorders, but as an essential step prior to use in disease studies, we set out to characterise their basal KP metabolome and investigate any changes to their overall phenotype in the liver and kidney, free of exogenous inflammatory stimuli. Both enzyme knockouts caused rapid alterations in accumulation of blood metabolite levels upstream of the affected enzyme, although downstream metabolite concentrations were surprisingly unaffected. KMO knockout elevated kynurenine, kynurenic acid and anthranilic acid, while QPRT knockout elevated quinolinic acid. Regardless of these significant metabolic alterations, histological examination of liver and kidney tissues, standard clinical blood chemistry and gross animal observations indicated no evidence of pathological changes in both the renal and hepatic systems. Our findings suggest that in a timeframe of 1–5 weeks and without evoked inflammation, robust homeostatic mechanisms can accommodate substantial fluctuations in KP metabolite concentrations in knockout mice without affecting renal or hepatic structure or function.

**Data availability statement:** The data underlying the results presented in the study are freely available from Figshare.com. The final figures can be accessed here - 10.6084/m9.figshare.29938676. The supporting raw data can be found here - 10.6084/m9.figshare.29938706.

**Funding:** The author(s) received no specific funding for this work.

**Competing interests:** The authors have declared that no competing interests exist.

## Introduction

The kynurenine pathway (KP) is the major metabolic route of the essential amino acid tryptophan (TRP) (Fig 1), and the *de novo* pathway for the biosynthesis of the cellular energy source nicotinamide adenine dinucleotide (NAD+). The liver is the primary site of approximately 90% of KP metabolism [1], while the kidneys have unique roles, able to perform both metabolite conversion and filtering/excreting metabolites circulating in the blood [2,3]. KP enzymes including KMO [4] and QPRT [5] are also highly expressed in the kidneys. KP activation triggering elevated expression of KP metabolic enzymes is well documented in a range of renal (e.g., acute kidney injury (AKI), chronic kidney disease (CKD), polycystic kidney disease (PKD)) [1,6,7] and hepatic (e.g., acute-on-chronic liver failure (ACLF) in cirrhosis, hepatic encephalopathy, and non-alcoholic fatty liver disease (NAFLD)) [8–11] disorders. This is not surprising, as the KP is potently activated by a range of inflammatory mediators and cytokines [12], while TRP depletion itself is inherently immunosuppressive, and several intermediary KP metabolites regulate the immune response by inducing immunosuppression and/or immune tolerance [13,14]. Modulating the KP is therefore a promising target for the generation of new therapeutics in renal and hepatic disease.

Insight into KP modulation for therapeutic utility has largely focused on effects relative to the immune system. However, many KP metabolites have diverse bioactive functions alongside their immunomodulatory/immunosuppressive roles, and changes in their physiological concentrations may negatively impact tissue health or function through non-immune mediated mechanisms. For example, kynurenic acid (KYNA) and quinolinic acid (QUIN) are antagonists and agonists respectively of the N-methyl-D-aspartate (NMDA) glutamate receptor [15–19]. In the central nervous system (CNS), chronic inflammation can preference continued QUIN production over it's clearance, where it accumulates in a variety of neurodegenerative diseases [20,21]. NMDA receptors are also expressed in the renal cortex and medulla, where they play a role in the regulation of renal blood flow and reabsorption [22]. The balance towards production of specific KP metabolites may also produce a pro-/anti-oxidative milieu, as QUIN is an established inducer of oxidative stress [23,24], while KYN [25], KYNA [23,26], and anthranilic acid (AA) [27] have antioxidant properties. QUIN is also a substrate for conversion in several steps into cellular cofactor NAD by enzyme QPRT; in essence a detoxification mechanism. The KP could also indirectly affect kidney function through its implicated role in haematopoiesis [28,29], which could influence renal perfusion through changes in blood volume and viscosity. Very few studies have investigated the effects of KP metabolites in healthy renal or hepatic systems. Proteinuria has been reported in mice lacking the KP enzyme kynurenine-3-monooxygenase (KMO) [30], although an alternative study of KMO knockout mice reported normal renal function under physiological conditions [4]. Collectively, these reports suggest that KP metabolites may affect the health of renal and hepatic tissues – a question this study aims to address.

We generated two novel mouse models which feature inducible adult-specific knockout of key KP enzymes KMO and QPRT. While existing studies have utilised global KP enzyme knockout mouse models to study pathway dysregulation free

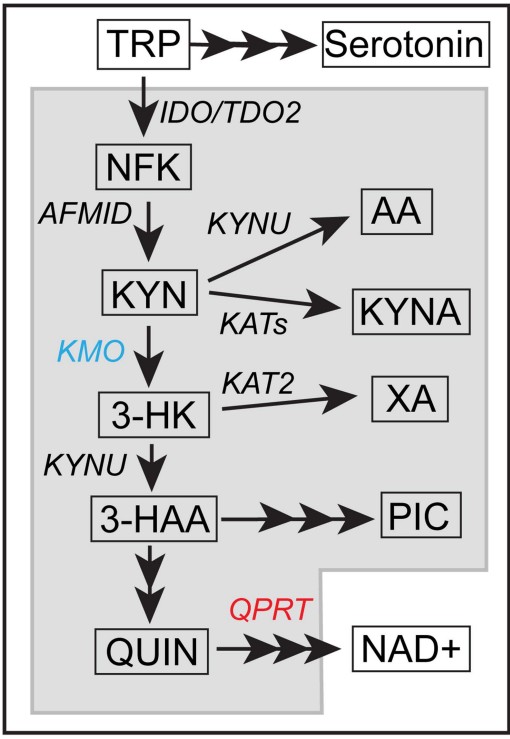

**Fig 1. The kynurenine pathway.** Schematic depicting tryptophan (TRP) metabolism through the kynurenine pathway (KP) showing metabolites in boxes and enzymes in italics. Multiple arrowheads indicate several steps are involved to produce the indicated metabolite. KMO (blue) and QPRT (red) are highlighted as indicated. Abbreviations: IDO, indoleamine-2,3-dioxygenase; TDO2, tryptophan-2,3-dioxygenase; NFK, n′-formylkynurenine; AFMID, arylformamidase; KYNU, kynureninase; KAT, kynurenine aminotransferase; KMO, kynurenine-3-monooxygenase; QPRT, quinolinate phosphoribosyl-transferase; KYN, kynurenine; KYNA, kynurenic acid; AA, anthranilic acid; 3-HK, 3-hydroxykynurenine; 3-HAA, 3-hydroxyanthranilic acid; XA, xan-thurenic acid; PIC, picolinic acid; QUIN, quinolinic acid; NAD +, nicotinamide adenine dinucleotide.

from inflammatory stimuli, a potential drawback of those is that the KP is involved in pregnancy, development, and the regulation of various stem cell niches [29,31]. While not causing embryonic lethality, KP enzyme knockout from birth may therefore produce developmental differences, physiological adaptations and tolerance that could potentially obscure the immediate effects of KP dysregulation in adults. Using an inducible knockout approach enables targeted disruption of KP activity in adulthood, providing an acute model to investigate the effects of enzyme loss from a precise induction timepoint, without lifelong compensatory changes.

## Methods

### Transgenic mice

Animal experiments were approved by the Garvan Institute of Medical Research and St Vincent's Hospital Animal Ethics Committee under project code 21/12, and were carried out in accordance with the Australian code of practice for the care and use of animals for scientific purposes. Mice were group housed (2–5 per cage) in individually ventilated cages, maintained on a 12-hour light/dark cycle and were provided ad libitum access to water and standard rodent chow. All mice were acclimatised for at least one week prior to experimental procedures and were randomly assigned to treatment groups while ensuring equal representation of sex and littermates across groups. The exact number of mice utilised in each experiment is detailed in subsequent methods sections and in figure legends. Details of methods of euthanasia and associated efforts to alleviate suffering are detailed in subsequent methods sections.

*Kmo^flox/flox* and *Qprt^flox/flox* mice were produced by the Mouse Engineering Garvan/ABR (MEGA) Facility (Moss Vale and Sydney, Australia) using CRISPR/Cas9 gene targeting in C57BL/6J mouse embryos following established molecular and animal husbandry techniques [32]. For *Kmo*, two single guide RNAs (sgRNAs) were employed that targeted Cas9 cleavage 416 bp 5' and 358 bp 3' of Exon 5 (AACTGTGTGTGACGGTA*ACATGG and CTTAGAGGTTACAGCCT*CGAGGG respectively, * = Cas9 cleavage site, protospacer-associated motif = PAM underlined). A homologous recombination (HR) substrate was synthesized in pUC57 plasmid (Genscript, Piscataway, NJ). This included a 3,719 bp insert corresponding to *Kmo* sequences from 1,416 bp 5' to 2,186 bp 3' of Exon 5, with 34 bp *loxP* sequences inserted at the two Cas9 cleavage sites. For *Qprt*, two single guide RNAs (sgRNAs) were employed that targeted Cas9 cleavage 398 bp 5' of Exon 2 and 85 bp 3' of Exon 4 (= 338 bp 3' of the STOP codon) (AACTGTGTGTGACGGTA*ACATGG and CGAACACTAACCTTCTG*GAGAGG respectively). The HR substrate plasmid in this case included a 5,026 bp insert corresponding to *Qprt* sequences from 2,398 bp 5' of Exon 2–1,085 bp 3' of Exon 4, with 34 bp *loxP* sequences inserted at the two Cas9 cleavage sites.

In each case, a solution consisting of the two sgRNAs (15 ng/μl each), purified double stranded HR template plasmid DNA (2 ng/μl) and full length, polyadenylated *S. pyogenes* Cas9 mRNA (30 ng/μl) was prepared and microinjected into the nucleus and cytoplasm of C57BL/6J zygotes. Microinjected embryos were cultured overnight and those that underwent cleavage introduced into pseudo-pregnant foster mothers. Pups were screened by PCR to detect homologous recombination of the two loxP sites into one of the *Kmo* or *Qprt* alleles. In each case a founder mouse was backcrossed to wild-type C57BL/6J mice and progeny inter-crossed to derive the homozygous *Kmo^flox/flox* and *Qprt^flox/flox* lines.

*B6.129-Gt(ROSA)26Sor^tm1(cre/ERT2)Tyj/J* [also called *R26-CreER^T2*; stock #008463, RRID:IMSR_JAX:008463 [33]] transgenic mice were purchased from Jackson Laboratories and maintained on a C57BL/6J background. *R26-CreER^T2* were crossed with *Kmo^flox/flox* or *Qprt^flox/flox* to generate experimental mice heterozygous for *R26-CreER^T2* and homozygous mice for either floxed *Qprt* or *Kmo* alleles. These are referred to in text as *R26-CreER^T2:: Qprt^flx/flx* and *R26-CreER^T2:: Kmo^flx/flx* transgenic mice.

## Genotyping

Genotyping was performed by the Garvan Molecular Genetics facility (NATA accredited ISO-17025). DNA was extracted from tail biopsies using the Magnetic Bead kit on the Chemagic™ 360 Nucleic Acid Extractor (Perkin Elmer). Transgenes were detected by touchdown realtime PCR with reactions conducted using Taq DNA Polymerase (MCLab, Molecular-cloning Laboratories Cat# TR-OEM) and PCR products analysed via high-resolution melt curve analysis (HRM) with the LightCycler 480 II system (Roche) using Syto9 dye and internal controls. PCRs were performed to detect *Cre* (For 5'- CCA TCA TCG AAG CTT CAC TGA AG -3', Rev 3'- GGA GTT TCA ATA CCC GAG ATC ATG C -5'), and *Cre* was confirmed as heterozygous or homozygous by a second PCR to detect wild type (WT) *Rosa26* (For 5'- CGT GAT CTG CAA CTC CAG TC -3', Rev 3'- CTG CTT ACA TAG TCT AAC TCG GAC -5'). *Kmo* was detected with three PCRs with primers designed to amplify the WT or floxed (FLX) gene, or a common (COM) sequence (PCR-1 *Kmo-WT* For 5'- GCC AAC ATA AGT CAT ATT G -3' and *Kmo-COM* Rev 3'- GCA CTT TTA AAA CTA TCT A -5'; PCR-2 *Kmo-WT* For 5'- GTT ACC GTC ACA CAC AGT T -3' and *Kmo-COM* Rev 3'- GCA CTT TTA AAA CTA TCT -5'; PCR-3 *Kmo-FLX* For 5'- GTA TAG CAT ACA TTA TAC G -3' and *Kmo-COM* Rev 3'- GCA CTT TTA AAA CTA TCT A -5'). Floxed *Qprt* was detected with a single PCR (For 5'- GCT AGA AAT TAA CAC CGC TC -3', Rev 3'- GAT GTC AGC TGC AGT TGC C -5'). All primers were used in a single reaction. The PCR program included: denaturation at 94°C for 10 seconds; 10 cycles of: 94°C for 10 seconds, 65°C to 55°C for 30 seconds with a touchdown of −1°C per cycle, and 60°C for 1 minute/kb; 30 cycles of: 94°C for 10 seconds, 55°C for 30 seconds, and 60°C for 1 minute/kb; completed with a final extension of 72°C for 3 minutes, and a hold at 4–10°C.

## Tamoxifen administration

Tamoxifen (Sigma Cat# T5648-5G) was dissolved in corn oil (Sigma Cat# C8267-500ML) by sonification, at a concentration of 40 mg/ml and administered to adult mice via oral gavage as four consecutive daily doses at 300 mg per kilogram

body weight. This regimen was selected based on published studies reporting high levels of recombination while being deemed a tolerable dose in adult mice, and a tolerable dose in our mice that did not have a noticeable adverse effects on mouse health or weight [34,35]. In the following results and discussion, treatments/experimental timepoints are represented relative to the first day of tamoxifen administration. Following administration, mice were monitored daily until experimental endpoint and sacrifice to ensure treatment and/or genetic knockout did not induce suffering.

## Blood collection, processing, and profiling

Mice were anesthetised by exposure to 5% isoflurane in $O_2$. Once nonresponsive, blood was collected via retro-orbital bleed using sterile glass capillaries (Hurst Scientific Cat# 11007311B-HS). Mice were monitored post-procedure to ensure lack of infection at the site, and recovery within expected timeframe. Hematology and analytical chemistry profiling was conducted on the same cohort of mice, consisting of a minimum of 9 mice per genotype and treatment, with equal representation of males and females. For hematology analysis, blood was collected in potassium EDTA coated tubes (Sarstedt Cat#41.1395.005) and analysed immediately after collection using the VetScan® HM5 Hematology Analyzer (Abaxis). For analytical chemistry profiling, blood was collected in lithium heparin coated tubes (BD Biosciences Cat# 365985) and analysed immediately after collection using the VetScan® VS2 Chemistry Analyser (Abaxis). To isolate blood plasma for profiling of blood metabolites via mass spectrometry, blood was drawn from an independent cohort of 6 mice per genotype and treatment, with attempts made to even match the number of males and females. Whole blood was collected in uncoated tubes, with 30 μL 0.5M pH 8.0 EDTA (Ambion Cat#AM9260G) added to inhibit blood coagulation. Plasma was separated from whole blood via centrifugation at 2500 RPM for 15 minutes at room temperature, and stored at −80°C prior to analysis. Mice used in these experiments were separate from mice used in tissue harvest/staining/capillary Western blot experiments.

## Mass spectrometry

Plasma KP metabolites were quantitated by a liquid chromatography-tandem mass spectrometry (LC-MS/MS) approach using a TSQ Vantage mass spectrometer (Thermo, Waltham, MA, USA) coupled to an integrated Vanquish (Thermo-Dionex, Thermo, Waltham, MA, USA) pump/autosampler system. Chromatographic separation was achieved using a Kinetex™ PFP column (150 mm, 2 mm, 1.7 μm, 100 Å, Phenomenex, Torrance, CA, USA). The method allowed reliable quantification of TRP, KYN, KYNA, AA, 3-hydroxykynurenine (3-HK), xanthurenic acid (XA), and serotonin. Details on sample preparation, chromatographic gradient, mass spectrometry parameters, have been reported in Bustamante et al (2022) [36]. GC/ECNI-MS analysis of QUIN and picolinic acid (PIC) in mouse plasma was performed using a 8890-5977B Agilent GC/MSD platform as described by Smythe et al (2002) [37]. Mice used in these experiments were separate from mice used in tissue harvest/staining/capillary Western blot experiments.

## Tissue collection

Mice were anesthetised by exposure to 5% isoflurane in $O_2$ until nonresponsive. Mice then received subcutaneous injection of the local anaesthetic bupivacaine hydrochloride (1 mg/kg at 0.025%) to reduce pain from the subsequent intraperitoneal injection of sodium pentobarbital (30 mg/kg); a method of terminal anaesthetisation. Once heart rate was sufficiently slowed and mice were on the verge of cardiac arrest, systemic blood was cleared via cardiac perfusion. To collect tissue for immunohistochemical and histological analysis, a cohort of mice (3 females and 3 males per genotype and treatment) were perfused with chilled PBS followed by 4% (w/v) paraformaldehyde (PFA; Sigma Cat# P6148-1KG) in PBS. Kidneys and livers were dissected, immersion fixed in 4% PFA overnight at 4°C, and transferred into PBS 0.02% sodium azide (Sigma Cat# S2002) for long term storage at 4°C. To collect tissue collected for western blot analysis, a second cohort of mice (3 females and 3 males per genotype and treatment) were perfused with chilled PBS only, and the kidneys and livers dissected, flash frozen in liquid nitrogen, and stored at −80°C.

## Automatic capillary western blotting

For protein extraction, tissue was homogenised in RIPA lysis buffer [50 mM Tris-HCl, 150 mM NaCl, 1% IGEPAL CA-630 (v/v; Sigma Cat#I8896), 0.5% sodium deoxycholate (w/v), 0.1% SDS, 5 mM EDTA, in MilliQ water, pH adjusted to 7.6] containing EDTA free protease (Roche Cat# 05892970001) and phosphatase (Roche Cat# 04906837001) inhibitor tablets, using a Q55 Sonicator® compact ultrasonic processor (Qsonica). Samples were centrifuged at 13,000 RPM for 10-minutes at 4°C, and supernatant was isolated and stored at −80°C. The concentration of extracted protein samples was determined by Detergent Compatible (DC) Protein Assay (Bio-Rad). Using technical triplicates, absorbance was measured using a FLUOstar Omega Microplate reader (BMG-Labtech), and protein concentrations determined with reference to a standard curve generated form Pre-Diluted Bovine Serum Albumin (ThermoFisher, Cat# 23208) across a dilution range of 0–2000 µg/ml.

Automated capillary western blotting was performed with the WES Automated Capillary-Based Size Sorting and Immunolabeling System (ProteinSimple). Samples were run as per the manufacturer's instructions using the 12-230kDa separation module (ProteinSimple, Cat# SM-W004). All primary antibodies were raised in rabbit and diluted to 1:50, and included KMO (Novus, Cat# NBP1−44263, RRID: AB_2133398), which was normalised to GAPDH (Abcam, Cat# ab181602, RRID: AB_2630358), and QPRT (LSBio, Cat# LS-B17002), which was normalised to β-tubulin (Abcam, Cat# ab6046, RRID: AB_2210370). Primary antibodies were detected using the Anti-Rabbit Detection Module (ProteinSimple, Cat# DM-001).

Data were analysed using the Compass for SW 4.1 software (ProteinSimple). The baseline chemiluminescence signal of each sample was adjusted to normalise to the relative background chemiluminescence signal; ensuring all samples were measured in identical conditions. The amount of each target protein was quantified by determining the area under the electropherogram peak at the associated molecular weight; KMO detected at ~53kDa, GAPDH at ~40kDa, QPRT at ~35kDa, and β-tubulin at ~54kDa. Peak detection settings were set for: Range (1–250), Baseline (Threshold 0.1, Window 400, Stiffness 0.1), and Peak Find (Threshold 10, Width 9, Gaussian Fit).

## Immunohistochemistry and staining

A standard protocol for Hematoxylin (Hematoxylin Harris non-toxic (acidified); Australian Biostain) and Eosin (Eosin Phloxine Alcoholic 1%; Australian Biostain) staining was performed on 4µm liver and kidney sections mounded on glass slides, using a Leica ST5010 Autostainer XL. For fluorescent immunohistochemistry, tissue was cryopreserved in 20% w/v sucrose in PBS at 4°C overnight and snap frozen in OCT (Scigen Cat# 4586). Immunohistochemistry was performed on free-floating 30µm cryosections. Tissue underwent heat induced antigen retrieval by incubating sections in Tris-EDTA buffer (10 mM Tris base, 1 mM EDTA solution, 0.05% Tween 20, pH 9.0) at 80°C for 20 minutes, were rinsed 3x10-minutes in PBS, and incubated for 1 hour at room temperature in blocking solution [PBS containing 0.1% v/v Triton X-100 (Sigma, Cat# T8787) and 10% heat-inactivated foetal bovine serum (FBS)]. Primary antibodies were diluted in blocking solution and incubated at 4°C overnight, rinsed 3x10 minutes in PBS and incubated with secondary antibodies diluted in blocking solution at 4°C overnight. Primary antibodies included rabbit anti-KMO (1:300; Abcam, Cat# ab83929, RRID: AB_1860298) and rabbit anti-QPRT (1:200; MyBioSource, Cat# MBS8307852). Secondary antibody used was a goat anti-rabbit conjugated to AlexaFluor-555 (1:1000; Thermo Fisher Scientific, Cat# A-21428, RRID: AB_2535849). Cell nuclei were labelled using 4′,6-diamidino-2-phenylindole (DAPI; 1:1000; Sigma Cat# D9542). Sections were mounted onto glass slides and cover slipped with fluorescent mounting medium (Sigma, Cat# F4680). Immunofluorescence images were captured on a Leica Thunder widefield microscope equipped with standard emission filters for DAPI and Alexa Fluor-555 and running LAS X v3.8 software. Images were processed for deblurring using instant computational clearing (ICC).

## Statistics

Statistical testing was performed in Prism (GraphPad, La Jolla, USA). Data distributions and normality were assessed using the Shapiro-Wilk test, and outliers were detected using the robust regression and outlier removal (ROUT) method. Normally distributed data were analysed using one-way ANOVA followed by the Tukey's multiple comparison post hoc

test. Data with unequal variance, assessed via the Brown-Forsythe test, were analysed by Welch's ANOVA followed by Dunnett's T3 multiple comparisons post-hoc test. Non-parametric data were analysed using the Kruskal-Wallis test followed by Dunn's multiple comparisons post-hoc test. Further information relating to statistical testing is presented in figure legends.

## Results

### KMO and QPRT are highly expressed in kidney proximal and distal tubules, and ubiquitously in the liver of control mice

The liver and kidneys are primary sites of KP metabolism, with key KP enzymes KMO and QPRT highly expressed by mammalian renal and hepatic cells [4,5,38–41]. Prior to knockout, we confirmed enzyme expression patterns by immunolabelling renal and hepatic tissue collected from untreated 6–8-week-old adult C57BL/6J mice. In the kidneys, the expected expression pattern was observed [41], with significant KMO and QPRT localised in the proximal and distal tubules (Fig 2A–D), in which expression has been previously confirmed in mouse [42] and human tubule epithelial cells [43]. For both enzymes, expression was also prominent in the kidney cortex and outer stripe of the medulla (OSOM), while QPRT was highly expressed in the inner stripe of the medulla (ISOM) where KMO expression was negligible. Neither enzyme was expressed to a significant degree in the glomeruli (Fig 2A,C boxed inset), renal calices, or renal pelvis. In the liver, both KMO and QPRT showed strong, uniform, ubiquitous expression within the tissue, representing high expression in liver hepatocytes (Fig 2E–H). Taken together, our findings confirm previous reports for the expression patterns of both KMO and QPRT in mouse liver and kidney tissue.

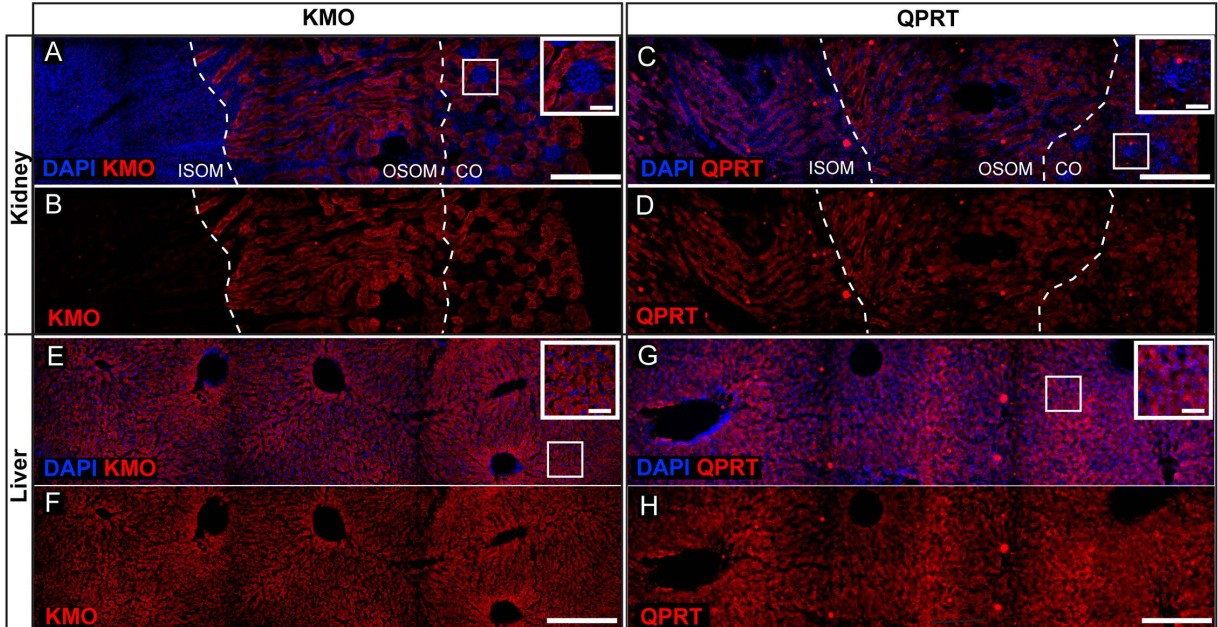

**Fig 2. KMO and QPRT are highly expressed in the normal mouse renal tubules and ubiquitously throughout the liver.** Representative fluorescent immunohistochemistry images of (A-D) axial kidney sections and (E-H) liver sections from untreated (n = 3) mice labelled for (A,B,E,F) (KMO; red) or (C,D,G,H) QPRT; red, and DAPI (blue). Dotted lines in A-D delineate the kidney cortex (CO), outer stripe of the medulla (OSOM), and inner stripe of the medulla (ISOM). White boxes are zoomed in the upper right of the same panel to show a zoomed inset and show examples of glomeruli (A,C) and hepatocytes (E,G). Scale bars represent 300μm; in inset boxes 50μm.

## Transgenic mice display efficient knockdown of KMO and QPRT in the kidneys, and knockout in the liver

Having confirmed KMO and QPRT basal expression patterns, we generated novel inducible *R26-CreER^T2^:: Kmo^flx/flx^* and *R26-CreER^T2^:: Qprt^flx/flx^* transgenic knockout mice, referred to from this point as KMO-iKO and QPRT-iKO respectively. Administration of tamoxifen to these mice produces global excision of the *Kmo* or *Qprt* genes, which facilitated the assessment of their loss on renal and hepatic function in an otherwise physiologically normal system.

Prior to evaluation of organ function, we assessed the efficiency of genetic knockout in the target tissues. 6–8-week-old KMO-iKO and QPRT-iKO experimental mice were administered tamoxifen and compared to controls, which included untreated KMO-iKO and QPRT-iKO mice, and C57BL/6J mice that were untreated, administered tamoxifen, or administered a vehicle treatment of corn oil. Liver and kidney tissue was collected from all groups 1 week after tamoxifen, and a separate cohort of tamoxifen-treated KMO-iKO and QPRT-iKO mice 5 weeks after tamoxifen. Enzyme expression was quantified using automated capillary western blotting, and qualitatively assessed using immunohistochemistry.

In KMO-iKO kidney tissue, capillary western blot analysis revealed a significant reduction of KMO protein to approximately one-third of control levels (decreased by $65.13 \pm 17.09\%$; mean ± SD) at 5 weeks post tamoxifen (Figs 3A–C and S1). Similarly, QPRT protein in the kidneys was reduced by $55.12 \pm 1.24\%$ (mean ± SD) in QPRT-iKO mice 5 weeks post-tamoxifen compared to physiological levels (Fig 3D–F). Efficient knockdown of both enzymes occurred in the liver, with KMO and QPRT knocked down by $51.90 \pm 13.61\%$ (mean ± SD) and $85.69 \pm 8.77\%$ (mean ± SD) respectively 1 week post-tamoxifen, with complete knockout was observed by 5 weeks (Fig 3G–L). Supporting the western blot findings, qualitative analysis of immunohistochemical labelling showed substantially reduced immunopositivity for renal and hepatic KMO and QPRT, respectively (Fig 3M–X). These findings validate the efficacy of enzyme knockout in both mouse strains.

## Upstream circulating KP metabolites are rapidly elevated following knockout of KMO or QPRT

Having validated and quantified the genetic knockout efficiency in KMO-iKO and QPRT-iKO mice, we profiled a panel of KP metabolites in blood plasma; a measure of circulating concentrations. As the liver is the primary site of TRP metabolism, and KMO and QPRT protein levels in the liver were both significantly reduced 1 week post-tamoxifen, plasma was analysed at 1 and 5 weeks post-tamoxifen in both mouse models and compared to tamoxifen-treated control mice. Measurements were determined for TRP, and KP metabolites KYN, KYNA, AA, 3-HK, XA, PIC, and QUIN (Fig 4A). Serotonin was also analysed, a TRP metabolite from the adjacent serotonin/melatonin pathway known to affect both hepatic and renal function [44–46]. KMO-iKO and QPRT-iKO strains were assessed independently, with comparisons limited to temporal changes within each model and to controls. A full list of metabolite concentrations is provided in S1 Table.

TRP and serotonin concentrations were comparable across all groups, indicating KMO or QPRT knockout does not perturb TRP usage (Fig 4B,C). KMO-iKO mice had significantly perturbed KYN degradation compared to control levels, with a ~2.2-fold KYN elevation 1 week post-tamoxifen and an elevation of ~28.8-fold by 5-weeks (Fig 4D). Direct KYN branch metabolites were also elevated, with KYNA levels increased by ~45-fold by 5 weeks (Fig 4E). While this elevation only showed a trend towards significance (p = 0.077) this is likely the result of the large variability in KYNA levels in the 5 week post-tamoxifen cohort, but even the lowest KYNA concentration in this group was ~6.5-fold higher than the highest concentration in the control cohort. Like KYNA, AA, the second direct branch metabolite of KYN, was increased ~15.8-fold 5 weeks after KMO knockout (Fig 4F). Surprisingly, KMO-iKO mice displayed no significant changes in concentrations of downstream KP metabolites 3-HK, XA, PIC, or QUIN, perhaps representing incomplete systemic KMO knockout, or that KYN degradation can bypass KMO (Fig 4G–J). As expected, QPRT-iKO mice had significantly elevated QUIN, with a ~6.6-fold increase by 5-weeks post-tamoxifen (Fig 4I), with no changes in any upstream metabolites. It should be noted that 3-HK, the direct product of KYN generated by KMO activity, was present at very low concentrations in controls (8 ± 9 nM; mean ± SD), with concentrations below the threshold of detection in some mice.

Given the described changes in metabolite concentrations, we also assessed the biologically significant ratios of KYN/TRP, often used as an indicator of how much TRP is being metabolised, and QUIN/KYNA due to their opposing effects on

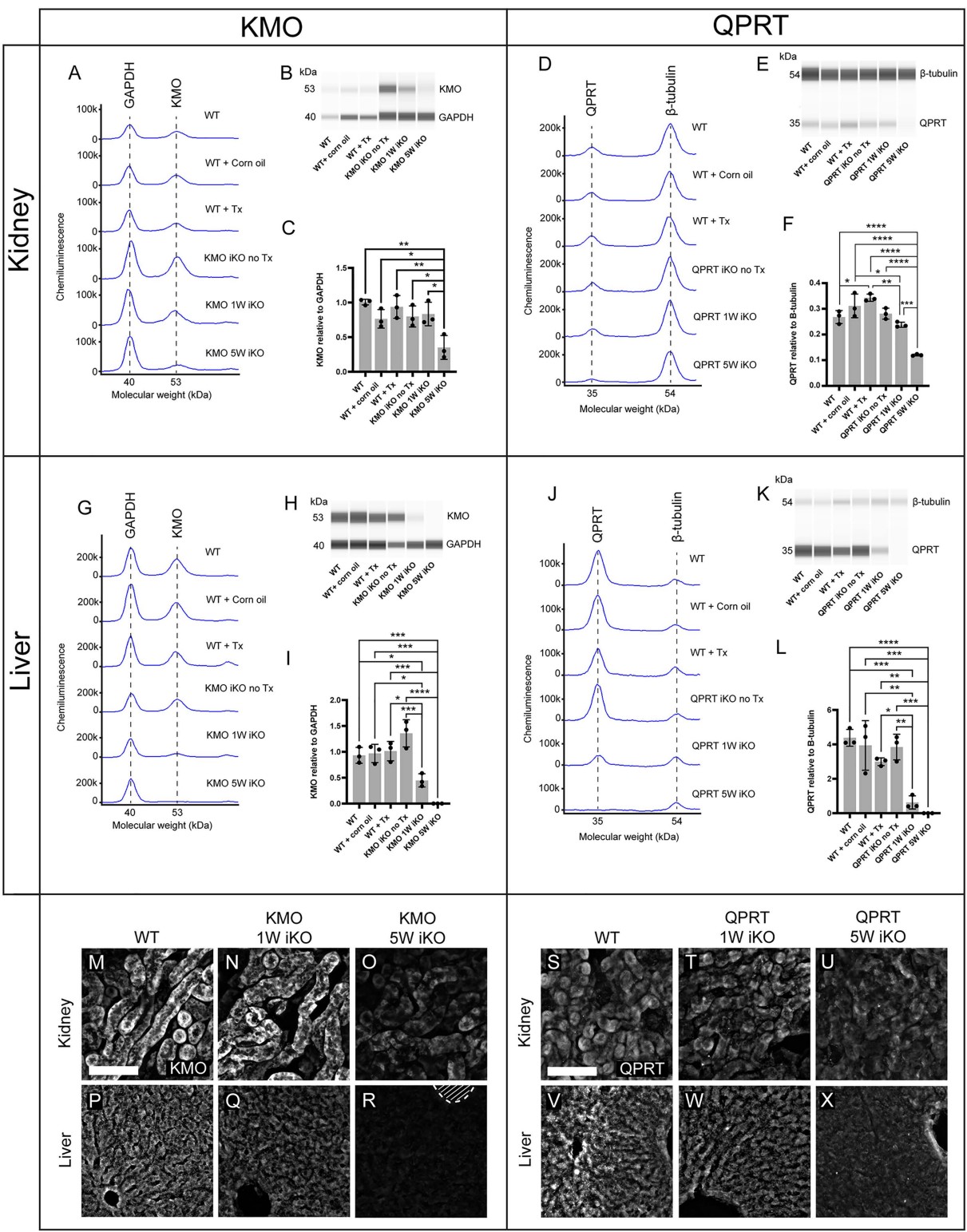

**Fig 3. KMO and QPRT protein is efficiently knocked down in the liver and kidneys of transgenic mice.** (A-L) Automated capillary western blot quantification of KMO protein in the kidney (A-C) and liver (G-I), and QPRT protein in the kidney (D-F) and liver (J-L), of *R26-CreER^T2:: Kmo^flx/flx* (KMO iKO) and *R26-CreER^T2:: Qprt^flx/flx* (QPRT iKO) mice respectively. KMO expression was normalised to GAPDH, and QPRT expression normalised to β-tubulin. KMO or QPRT were quantified at 1 week (1W) or 5 weeks (5W) post-tamoxifen (Tx) and compared to control groups of untreated, Tx treated,

or corn oil treated (vehicle control) C57BL/6 (wild type; WT) mice, and untreated KMO/QPRT iKO mice. Protein levels are illustrated by representative electropherogram peaks (A,D,G,J) and computer-generated western blots (B,E,H,K). Bar graphs depict the relative quantification, comparing experimental and control groups, for KMO protein in the (C) kidney (1-way ANOVA: $F_{(5,12)} = 7.4$, $p = 0.002$) and (I) liver (1-way ANOVA: $F_{(5,12)} = 24.0$, $p < 0.0001$) of KMO iKO mice, and QPRT protein in the (F) kidney (1-way ANOVA: $F_{(5,12)} = 31.00$, $p < 0.0001$) and (L) liver (1-way ANOVA: $F_{(5,12)} = 20.33$, $p < 0.0001$) of QPRT iKO mice. (M-R) Representative fluorescent immunohistochemistry images showing the knockdown of KMO (white) between untreated WT mice, and KMO iKO mice 1 week or 5 weeks post-Tx, in (M-O) the kidney cortex in axial kidney sections, and (P-Q) liver sections. (S-X) Representative fluorescent immunohistochemistry images showing the knockdown of QPRT (greyscale) between untreated WT mice, and QPRT iKO mice 1 week or 5 weeks post-Tx, in (S-U) the kidney cortex in axial kidney sections, and (V-X) liver sections. Delineated area in R represents hepatic central vein. Scale bars represent 140 µm. Data are presented as mean ± SD. Closed circles represent individual mice (n = 3 per group). The significance of differences between groups was evaluated using Tukey's post-hoc tests, indicated as *$p < 0.05$, **$p < 0.01$, ***$p < 0.001$, ****$p < 0.0001$.

NMDA receptors. As expected, the KYN*10/TRP ratio in KMO-iKO mice knockout trended towards a statistical increase by 1 week ($p = 0.071$; ~1.8-fold) and was significantly increased by ~22.6-fold by 5 weeks post knockout (Fig 4K). The QUIN/KYNA ratio was also significantly reduced by ~27.6-fold by 5 weeks. The opposite effect was observed in QPRT-iKO mice, where the QUIN/KYNA ratio was significantly elevated by ~2.8- and ~8.7-fold respectively 1 and 5 weeks after QPRT knockout. Together these data suggest that the KP is significantly perturbed in these mice and facilitate the study of altered circulating metabolite concentrations on organ function.

### Kidney and liver histology was unaffected in KO mice

Regardless of the significant KP metabolic alterations described above, no gross phenotypic changes were observed in either model. Mice displayed normal grooming, movement and activity levels, weight maintenance, normal hindlimb extension and leg splay reflexes during tail suspension, normal fur condition, and showed no overt signs of pain or distress. To assess any impact of KMO or QPRT knockout on renal and hepatic tissue, kidney and liver was collected from control, KMO-iKO, and QPRT-iKO mice 5 weeks post tamoxifen administration, and histologically analysed using hematoxylin and eosin staining.

Kidney tissue displayed no overt signs of pathology in either knockout model (Fig 5A–F). Gross renal architecture remained intact with no abnormalities or distortions in the renal cortex or medulla, and these areas remained clearly demarcated. There was no apparent gross change in the glomeruli and tubules, which appeared normal with no evidence of inflammation or fibrosis. Cellular-level microscope analysis showed no evidence of significant inflammatory infiltrate or changes in total cell density (hyper- or hypo-cellularity) in any region, or changes in nuclear structure such as fragmentation that could indicate cell death. Glomeruli morphology appeared normal, and the glomerular capillaries were intact. The renal tubules appeared healthy with normal epithelial lining free of sloughing or vacuolization, and there was no evidence of atrophy, degeneration or acute damage, or changes in the tubule brush border. We did observe a subtle but consistent increase in renal tubule dilation, but this was not to the extent to suggest pathological changes.

Similarly, no significant histopathological differences were observed in the liver (Fig 5G–L), aligning with a previous report of hepatocyte-specific KMO knockout mouse [47]. Tissue exhibited typical hepatic architecture, and hepatocytes retained normal morphology with a uniform cellular structure and no visible signs of cellular distress such as nuclear pleomorphism or cytoplasmic abnormalities. Minor differences in hepatocyte glycogen stores were observed, presenting as clear cytoplasmic vacuoles, but this appeared to represent normal physiological variation. Sinusoidal spaces remained unaltered, suggesting normal blood flow and no signs of inflammation. Minimal inflammation was observed, with a small number of immune cell aggregates across mouse lines, indicating the broad absence of overt inflammatory processes affecting the liver tissue.

These findings suggest that knockout of KMO or QPRT, and the associated elevations of key KP metabolites, does not acutely alter renal or hepatic tissue histology.

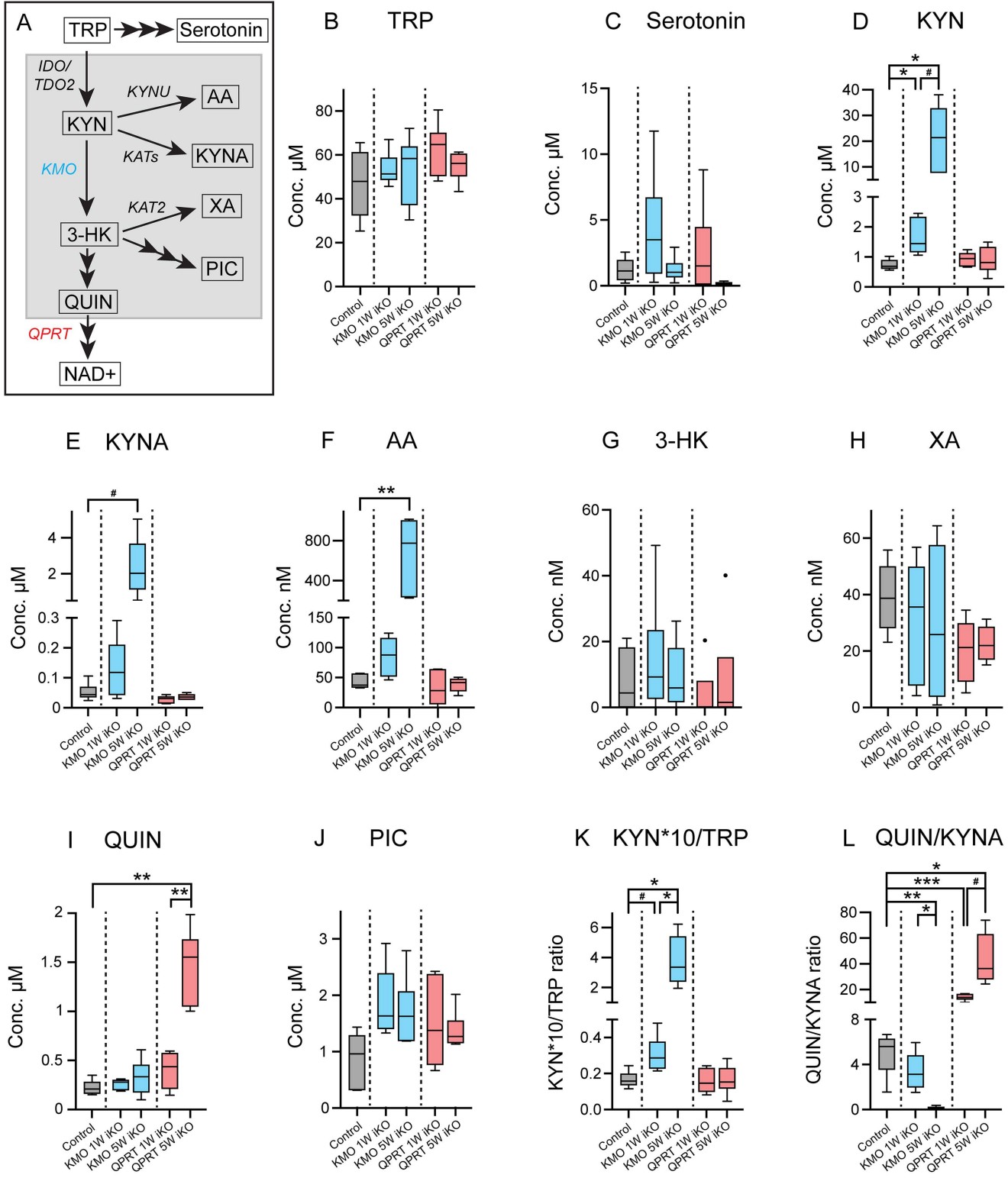

**Fig 4. Measurement of KP metabolites in blood plasma revealed elevated KYN, KYNA and AA following KMO knockout, and elevated QUIN following QPRT knockout.** Schematic depicting the metabolism of TRP through the KP (grey box) or to serotonin. Metabolites are shown in boxes with arrows pointing to subsequent products, and enzymes are shown in italics. Multiple arrowheads indicated steps required to generate metabolite.

KMO is the target of knockout in *R26-CreER*[T2]*:: Kmo*[flx/flx] (KMO iKO) mice, and is shown in blue. QPRT is the target of knockout in *R26-CreER*[T2]*:: Qprt*[flx/flx] (QPRT iKO) mice, and is shown in red. These colours are used in graphs B-L to depict changes in metabolites following knockout of the associated enzyme. (B-J) Tukey box plots showing comparisons between the concentration of metabolites measured in the blood plasma of tamoxifen treated C57BL/6J control mice, and KMO iKO and QPRT iKO mice at 1-week (1W) or 5-weeks (5W) post-tamoxifen (n = 6 per group). (B) TRP (1-way ANOVA: $F_{(4,15)} = 1.280$, $p = 0.304$), (C) serotonin (Kruskal-Wallis test: $H = 11.36$, $p = 0.023$), (D) kynurenine (KYN; Welch's ANOVA: $F_{(4,11.68)} = 6.654$, $p = 0.005$), (E) kynurenic acid (KYNA; Welch's ANOVA: $F_{(4,11.57)} = 4.816$, $p = 0.016$), (F) anthranilic acid (AA; Kruskal-Wallis test: $H = 19.38$, $p = 0.001$), (G) 3-hydroxykynurenine (3-HK; Kruskal-Wallis test: $H = 3.495$, $p = 0.479$), (H) xanthurenic acid (XA; Welch's ANOVA: $F_{(4,11.02)} = 2.316$, $p = 0.406$), (I) quinolinic acid (QUIN; Welch's ANOVA: $F_{(4,11.56)} = 14.26$, $p = 0.0002$), and (J) picolinic acid (PIC; Kruskal-Wallis test: $H = 9.011$, $p = 0.061$). (K-L) Comparisons of the biologically significant ratios between KP metabolites (K) KYN/TRP (Welch's ANOVA: $F_{(4,11.95)} = 8.783$, $p = 0.002$), and (L) QUIN/KYNA (Welch's ANOVA: $F_{(4,10.05)} = 55.09$, $p < 0.0001$). To determine the KYN/TRP ratio, KYN concentration was multiplied by a factor of 10 as physiological concentrations of TRP are in µM while KYN are in nM. In Tukey box plots, the box depicts interquartile range (IQR), line represents the median, bars extend to the furthest data points within 1.5 times the IQR from the quartiles, and closed circles represent statistical outliers. The significance of differences between groups was evaluated using select comparison post-hoc testing between within each iKO strain and in comparison to controls, with no comparison between KMO iKO and QPRT iKO mice. Post-hoc tests included Tukey's (B), Dunnett's T3 (C,D,G,H,K,L), or Dunn's (E,F,I,J), indicated as #p = < 0.08 *p < 0.05, **p < 0.01, ***p < 0.001. Abbreviations: IDO, indoleamine-2,3-dioxygenase; TDO, tryptophan-2,3-dioxygenase; KMO, kynurenine-3-monooxygenase; KYNU, kynureninase; KAT, kynurenine aminotransferase- (isoforms 1, 2 and 3); NAD +, nicotinamide adenine dinucleotide; QPRT, quinolate phosphoribosyltransferase, TRP, tryptophan.

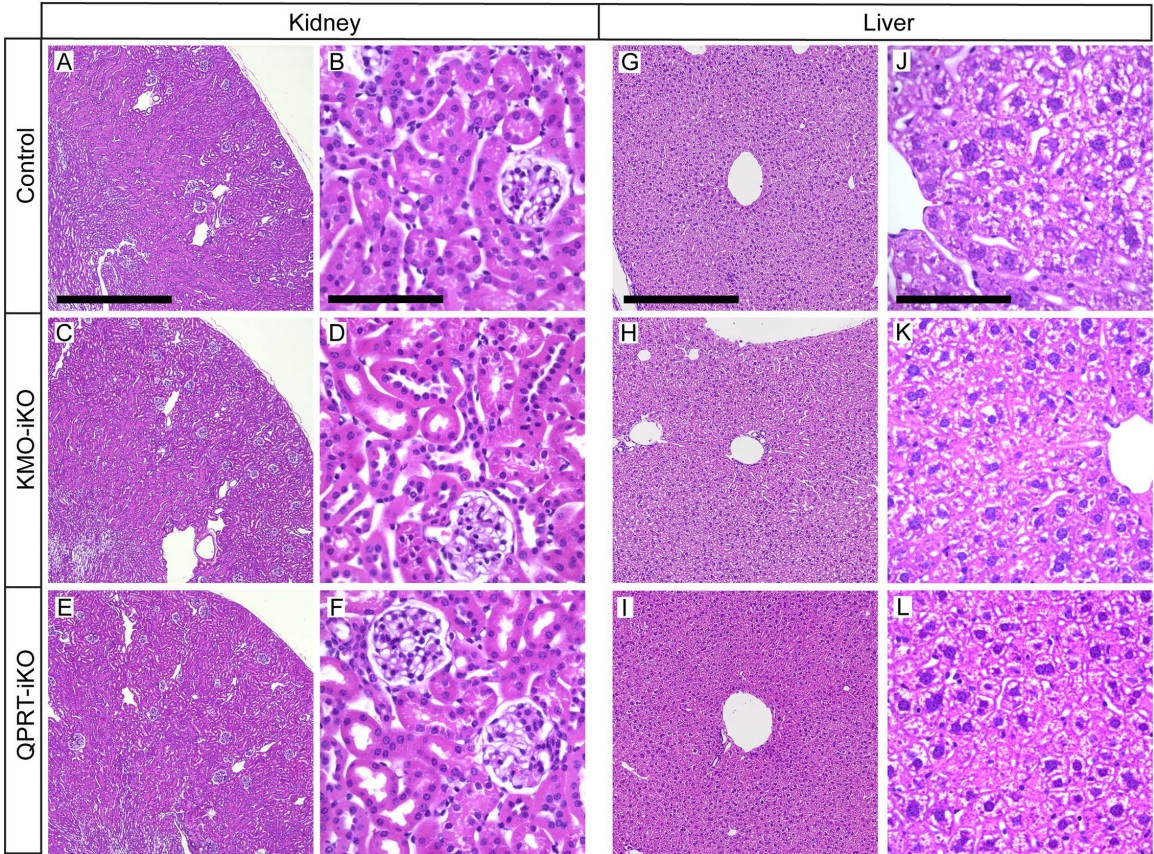

**Fig 5. Histological assessment of kidney and liver tissue reveals no overt changes following knockout of KMO or QPRT.** Representative images of H&E stained tissue sections from the (A-F) kidney and (G-I) liver of C57BL/6J, *R26-CreER*[T2]*:: Kmo*[flx/flx] (KMO-iKO) and *R26-CreER*[T2]*:: Qprt*[flx/flx] (QPRT-iKO) mice, collected 5-week post tamoxifen administration (n = 3 per group). Kidney images display (A,C,E) the architecture of the renal cortex and medulla, and (B,D,F) the detail cellular structure of the glomeruli, proximal tubules, and distal tubules in the renal cortex. (G,H,I) representative images of the liver, showing a typical central portal vein and ordered arrangement of surrounding hepatocytes. Scale bars represent 800µm (A,C,E), 100µm (B,D,F,J,K,L), 350µm (G-I).

## Blood hematology and chemistry profiles were unchanged by KMO or QPRT knockout

While histological analysis revealed no evidence of gross tissue/cellular changes in the liver or kidneys of KMO-iKO or QPRT-iKO mice, we also assessed organ function via analytical chemistry blood testing and hematology analysis 5 weeks after induction of knockout. Values were compared to control mice, and all groups were compared to publicly available baseline datasets for C57BL/6J mice from mouse suppliers (Charles River Laboratories and The Jackson Laboratory) as a benchmark of normal values or ranges. Values and statistical comparisons of selected relevant tests are presented in Table 1, with additional tests presented in S2 Table.

Supporting our histological findings, analytical chemistry testing revealed no evidence of renal or hepatic dysfunction. Assessing renal function, blood urea nitrogen (BUN) was unaffected, and while phosphorus levels were significantly elevated in both KMO-iKO ($p = 0.041$) and QPRT-iKO ($p = 0.002$) mice, concentrations were not outside of normal physiological ranges. Hepatic function indicators albumin, alkaline phosphatase, alanine aminotransferase, total bilirubin, and

**Table 1. Analytical chemistry and hematology testing results for KMO and QPRT-iKO mice.**

| | | Group | | | | |
|---|---|---|---|---|---|---|
| | | Control (C57BL/6J) | KMO-iKO | | QPRT-iKO | |
| | | n = 9 (F = 4, M = 5) | n = 9 (F = 4, M = 5) | | n = 10 (F = 7, M = 3) | |
| **Clinical chemistry** | | Value (mean ± SD) | Value (mean ± SD) | Difference from control | Value (mean ± SD) | Difference from control |
| Albumin | g/L | 35.00 ± 1.80 | 34.22 ± 2.39 | NS | 37.70 ± 1.16 | ** |
| Alkaline phosphatase | U/L | 97.78 ± 29.90 | 95.89 ± 30.46 | NS | 99.90 ± 18.92 | NS |
| Alanine aminotransferase | U/L | 28.67 ± 12.45 | 39.50 ± 25.09 | NS | 30.00 ± 13.64 | NS |
| Total bilirubin | umol/L | 6.11 ± 0.60 | 5.89 ± 0.93 | NS | 6.00 ± 0.47 | NS |
| Blood urea nitrogen | mmol/L | 10.48 ± 2.63 | 8.38 ± 0.68 | NS | 10.20 ± 1.07 | NS |
| Calcium | mmol/L | 2.43 ± 0.12 | 2.37 ± 0.04 | NS | 2.40 ± 0.09 | NS |
| Phosphorus | mmol/L | 2.15 ± 0.15 | 2.52 ± 0.47 | * | 2.44 ± 0.11 | ** |
| Glucose | mmol/L | 14.99 ± 1.90 | 15.81 ± 3.10 | NS | 14.23 ± 1.40 | NS |
| Sodium | mmol/L | 149.9 ± 1.05 | 149.1 ± 2.15 | NS | 150.3 ± 2.50 | NS |
| Potassium | mmol/L | 4.93 ± 0.32 | 5.44 ± 0.47 | NS | 5.21 ± 0.72 | NS |
| Total protein | g/L | 47.44 ± 3.09 | 47.33 ± 2.35 | NS | 50.40 ± 2.91 | # |
| Globulin | g/L | 12.22 ± 1.72 | 13.11 ± 2.57 | NS | 12.40 ± 1.90 | NS |
| **Hematology** | | n = 10 (F = 4, M = 6) | n = 10 (F = 5, M = 5) | | n = 12 (F = 9, M = 3) | |
| White blood cell count (WBC) | 10^9/L | 10.24 ± 2.26 | 7.12 ± 2.84 | ** | 8.12 ± 1.60 | # |
| Lymphocyte count | 10^9/L | 8.47 ± 3.20 | 6.54 ± 2.87 | NS | 7.40 ± 1.39 | NS |
| Monocyte count | 10^9/L | 0.19 ± 0.10 | 0.17 ± 0.14 | NS | 0.32 ± 0.36 | NS |
| Neutrophil count | 10^9/L | 0.58 ± 0.37 | 0.41 ± 0.29 | NS | 0.37 ± 0.20 | NS |
| Percent lymphocytes | % | 92.22 ± 4.48 | 90.07 ± 9.21 | NS | 92.22 ± 4.84 | NS |
| Percent monocytes | % | 1.88 ± 0.94 | 2.54 ± 1.68 | NS | 3.66 ± 3.35 | NS |
| Percent neutrophils | % | 5.88 ± 3.87 | 7.37 ± 8.04 | NS | 4.57 ± 1.99 | NS |
| Red blood cell count (RBC) | 10^12/L | 8.77 ± 1.37 | 9.70 ± 0.53 | NS | 9.17 ± 0.84 | NS |
| Hemoglobin | g/dL | 11.42 ± 1.89 | 12.51 ± 1.02 | NS | 12.64 ± 1.23 | NS |
| Hematocrit | % | 38.97 ± 5.68 | 42.39 ± 2.68 | NS | 39.96 ± 3.65 | NS |
| Platelet count | 10^9/L | 360.20 ± 155.66 | 421.30 ± 204.05 | NS | 306.42 ± 138.43 | NS |

Abbreviations: KMO-iKO, *R26-CreER*[T2]:: *Kmo*[flx/flx]; QPRT-iKO, *R26-CreER*[T2]:: *Qprt*[flx/flx]; #p = <0.08 *p < 0.05, **p < 0.01 versus control.

globulin were unchanged other than a slight increase in blood albumin in QPRT-iKO mice (p = 0.006) that remained within an expected physiological range. To assess if altered hydration status may mask a reduction in globulin, we examined serum globulin concentrations alongside calculated albumin-to-globulin (A/G) ratios. Both globulin and A/G ratio were similar across groups (Control: 12.22 ± 1.72 g/L, A/G = 2.87; KMO-iKO: 13.11 ± 2.57 g/L, A/G = 2.61; QPRT-iKO: 12.40 ± 1.90 g/L, A/G = 3.04), and together with stable hematocrit, hemoglobin, and sodium, this analysis suggested globulin levels were unchanged. No significant changes were seen in total protein, calcium, glucose, potassium, or sodium levels.

Comprehensive hematology analysis also revealed no evidence of pathological changes in haematopoiesis or blood cell parameters in response to enzyme knockout. Total white blood cell counts were significantly reduced in KMO-iKO mice (p = 0.008) and trended towards a significant reduction in QPRT-iKO mice (p = 0.066), potentially suggesting immunosuppression. However, these decreases did not drop below previous reports of normal physiological counts, and did not translate to changes in counts or percentages of lymphocytes, monocytes, or neutrophils. Platelet counts were slightly elevated compared to baseline reference datasets, but this was also observed in our control cohort with no difference observed in either knockout mouse model.

Together, these data support the histological analysis, showing no renal or hepatic dysfunction following acute knockout of the KMO or QPRT enzymes.

## Discussion

The liver and kidneys are major sites of KP activity, and dysregulation of the KP is well documented in renal and hepatic disorders. However, the impact of acute changes in bioactive KP metabolite concentrations on renal and hepatic function, outside of their immunomodulatory effects, is poorly understood. We investigated if genetic deletion of enzymes KMO or QPRT in adult mice, causing elevations of key KP metabolites in a timeframe of 5 weeks did not precipitate renal or hepatic dysfunction in an otherwise physiologically healthy system free from inflammation.

### Generation of two novel inducible KP enzyme knockout mouse lines

We generated novel inducible QPRT and KMO knockout mice, an improvement over existing global null knockout mice which might reflect biases like potential developmental physiological adaptations or compensatory changes by adult. These mice are poised to be powerful tools in the study of our understanding of each of these enzyme's function, since knockout can be triggered in a temporal-specific manner. We showed that inducible genetic manipulation produced rapid accumulation of direct upstream and branch KP metabolites.

### KMO as a midpoint KP enzyme with higher activity than branchpoint enzymes

In KMO-iKO mice we observed rapid increases in circulating concentrations of direct upstream metabolite KYN, which was alternatively channelled into the production of metabolites KYNA and AA, largely aligning with previous reports [4,30,48–51]. Considering the accumulation of these branch metabolites, it is evident the majority of KYN undergoes hydroxylation by KMO in preference to metabolism by AA-converting enzyme kynureninase (KYNU), or KYNA converting enzyme kynurenine aminotransferase (KAT; isoforms 1–3). This may represent both higher expression of KMO and/or a higher affinity of KYN by KMO, ensuring that KYN metabolism is primarily directed to produce downstream metabolites rather than metabolism by alternative branchpoint enzymes (Fig 1). Certainly, activity of KMO in the mouse liver is reported to be around double that of KYNU utilising KYN as a substrate, and >30-fold higher than the activity of KAT-2 [49]. The rapid accumulation of KYN also suggests that the activity of both KYNU and the KATs combined is insufficient to metabolise KYN more rapidly than it accumulates, even at basal metabolic rates. This underscores KMO's pivotal role in the bi-directional control of metabolite levels, particularly in facilitating the conversion of KYN to 3-HK, which is a key step towards the synthesis of downstream metabolites such as XA and QUIN.

We did note that KMO-iKO mice did not have the expected reductions in downstream metabolites 3-HK, XA, PIC, or QUIN, which have previously been reported in KMO *null* mice [4,48,49,51]. This is likely due to incomplete knockout in our model (Fig 3), which probably retains sufficient KMO in certain cells to maintain low levels of downstream circulating metabolites. While this is presumed to be largely in hepatocytes, which account for the majority of systemic KMO activity [47], other cell populations in other tissues may also contribute. However, previous reports of downstream metabolites being detected at very low levels in KMO *null* mice suggests the existence of alternative degradation pathways. Interestingly, one study has reported a decrease in serum AA in KMO null mice [30], which the authors suggested this may represent the direct conversion of AA to 3-hydroxyanthranilic acid (3-HAA), the metabolite downstream of 3-HK. This conversion is believed to be catalysed by the oxidoreductase enzyme anthranilate 3-monooxygenase (FAD), which is not reported to be expressed by mammals [52]. Considering this is the only existing report of AA depletion in KMO null mice, we find it more likely that non-specific hydroxylases may convert small amount of AA to 3-HAA, and KYN to 3-HK to bypass KMO. This has been described for 3-HAA [53]. While 3-HAA was not in our analysis panel, as a QUIN precursor this would align with the finding of increased QUIN, especially at 5 weeks. Overall, these data highlight a regulation mechanism in these organs that retains a degree of homeostasis even in response to significant genetic perturbation to KP enzyme expression and metabolite levels.

### QPRT is an important regulator of circulating QUIN levels; relevance to NAD production, especially in the liver

Aligning with previous studies utilising non-inducible QPRT knockout strains [54–57], QPRT knockout selectively elevated QUIN, with no feedback mechanisms affecting upstream enzyme activity. Eliminating de novo NAD production from QUIN in the QPRT knockout animals may have shifted the emphasis towards dietary nicotinamide riboside and niacin/vitamin B3 as the source of NAD precursor in the liver.

### Blood and clinical chemistry showed only subtle changes in KO mice

Regardless of KP dysregulation, QPRT-iKO and KMO-iKO were largely unaffected across a range of clinical chemistry and hematology parameters, although some subtle changes warrant discussion. We note that these differences were drawn in comparison to our control cohort, and our findings must also be considered in respect to publicly available online databases of C57BL6 mice bloodwork (Charles River Laboratories; The Jackson Laboratory). While some of our values extend outside of physiologically normal ranges for one dataset, they remain inside for another, demonstrating the intrinsic variability which may be influenced by factors such as mouse housing conditions, diet, microbiota and even geographical location [58]. We therefore emphasise consideration of our findings in the context of our control cohort, which normalises the effect of tamoxifen treatment and the specific holding conditions of our mouse housing facility (pathogen-free and routinely monitored).

Comprehensive hematology analysis showed a significant decrease in total white blood cell counts in KMO-iKO mice. Activation of the KP is observed in a range of conditions with low blood cell count, termed cytopenias, and KYN and KYNA inhibit haematopoiesis [28]. However, considering we saw no changes in red blood cell number, or in the number of specific white blood cell populations (lymphocytes, monocytes, and neutrophils), and ranges in all groups remained well within previously reported range, we conclude that observed differences are unlikely to be biologically significant. It must also be highlighted that blood cells (leukocytes and platelets) can contribute to KP metabolism, although this activity is potently regulated by inflammatory signalling molecules and cytokines (e.g. interferon gamma) [59–61]. Their contribution to circulating KP metabolite concentrations basally is negligible (in contrast to renal and hepatic metabolism), as highlighted by our previous studies of human monocytes, lymphocytes and dendritic cells [61,62].

For clinical chemistry analysis, QPRT-iKO mice had significant increases in blood albumin and phosphorus, and a trend towards a significant increase in total protein. KMO-iKO mice also presented a slight increase in phosphorus, the only effect observed in this model. A primary reason for small elevations of albumin and total protein is dehydration [63], which

could also contribute to increased phosphorus. Excess QUIN may alter renal reabsorption through its agonism of NMDA receptors localised in the glomeruli and renal tubules [64,65], producing increase glomerular filtration rate (GFR) [66] and excessive water loss to urine. While a large increase in the excretion of water may be expected to change the concentration of circulating sodium, potassium or calcium ions, which we did not observed, a subtle increase in ion absorption would likely be compensated for by the bodies robust regulatory mechanisms to maintain electrolyte balance [67,68], which could be assessed through urine output. However, increased blood content of BUN and globulin would be expected following hemoconcentration [63], which we did not observe. While dehydration can increase the concentration of plasma proteins through hemoconcentration and mask reductions in globulins, mainly γ-globulins, our findings do not support this mechanism. Globulin levels and albumin-to-globulin ratios were unchanged, and markers of hydration status, including hematocrit, hemoglobin, and sodium, were stable. γ-globulins are primarily produced by B lymphocyte derived plasma cells, and as our lymphocyte counts were preserved in both knockout models, therefore a substantial loss of γ-globulin production is unlikely. Unfortunately, our measurement protocols were unable to accurately detect creatine at levels normally observed in mice, which could have provided additional information on GFR. It is possible that these data indicate subtle changes that may compound when dysregulation is chronic, a target for future studies.

In conclusion, our study demonstrates that acute dysregulation of the KP, via knockout of KMO or QPRT, does not significantly alter the physiological functions in the liver and kidneys despite significant elevations in the concentration of certain bioactive KP metabolites. This suggests a robustness in cells in these organs, that can accommodate substantial fluctuations in KP metabolite concentrations without immediately affecting their health or function. At least acutely, worsening outcomes from KP modulation in animal models of renal or hepatic disease are likely from immune-mediated mechanisms, where high levels of inflammatory cytokines trigger elevated KP gene transcription. However, we observed subtle changes in blood chemistry and hematology profiles, raising the potential for long-term consequences or more pronounced effects in a chronic setting or under pathological conditions where the KP is upregulated by inflammatory stimuli.

## Supporting information

**S1 Table. Concentrations of KP metabolites in blood plasma of control, and KMO or QPRT knockout mice.**
Abbreviations: KMO, kynurenine 3-monooxygenase; QPRT, quinolinate phosphoribosyltransferase; iKO, inducible knockout; +7/35D, 7/35 days post tamoxifen; TRP, tryptophan; KYN, kynurenine; KYNA, kynurenic acid; AA, anthranilic acid; 3-HK, 3-hydroxykynurenine; XA, xanthurenic acid; PIC, picolinic acid; QUIN, quinolinic acid.
(DOCX)

**S2 Table. Additional analytical chemistry and hematology tests.** *p < 0.05, **p < 0.01.
(DOCX)

**S1 Fig. Computer-generated WES "blot" outputs (uncropped and unadjusted) for KMO and QPRT in liver and kidney tissue.** Computer-generated blot images produced by the ProteinSimple WES automated capillary. electrophoresis system. These images are not traditional Western blots. Instead, they are software-generated representations created from the underlying electropherogram peak data to provide a format familiar to readers. The raw electropherogram traces were the actual data analyzed in this study (see Supporting Information). The images shown here are uncropped, unadjusted, and presented exactly as exported from the WES software.
(PDF)

## Acknowledgments

We acknowledge the Mouse Engineering Garvan/ABR (MEGA) Facility (Moss Vale and Sydney, Australia) for the generation of inducible transgenic mouse strains. We acknowledge the Garvan Institute Histopathology and Biospecimen facility's service for assistance with tissue processing, sectioning and histological staining. We acknowledge Garvan Biological

Testing Facility (BTF) and Australian BioResources Pty Ltd (ABR) animal facilities staff for their expertise and assistance with mouse colony maintenance, and Garvan Molecular Genetics (GMG) facility for assistance with genotyping. We acknowledge the access provided to the Leica Thunder microscope (part of the St. Vincent's Centre for Applied Medical Research (AMR) Live Imaging Core Facility (LIF)) for imaging data acquisition in this study. We acknowledge the technical assistance and access to equipment required to undertake the metabolomic analyses at the Bioanalytical Mass Spectrometry Facility (BMSF) at the Mark Wainwright Analytical Centre (MWAC), University of New South Wales, Sydney. We acknowledge the helpful comments on the liver and kidney hematoxylin and eosin histopathology from Dr. Dino Premilovac, University of Tasmania, Tasmanian School of Medicine, Hobart, Australia, and liver immunostaining by A/Prof. Patrick Bertolino, Centenary Institute, Sydney, Australia.

## Author contributions

**Conceptualization:** Benjamin S. Summers, Luke Milham, Sonia Bustamante, Peggy Rentsch, Gayathri Sundaram, Bruce J. Brew.

**Data curation:** Benjamin S. Summers, Krishan Gondal, Gayathri Sundaram, Bruce J. Brew.

**Formal analysis:** Benjamin S. Summers, Luke Milham, Sonia Bustamante, Krishan Gondal, Michael D. Lovelace.

**Funding acquisition:** Bruce J. Brew.

**Investigation:** Benjamin S. Summers, Luke Milham, Sonia Bustamante, Krishan Gondal, Peggy Rentsch, Gayathri Sundaram.

**Methodology:** Benjamin S. Summers, Luke Milham, Sonia Bustamante, Krishan Gondal, Peggy Rentsch, Gayathri Sundaram, Bryce Vissel.

**Project administration:** Michael D. Lovelace, Bruce J. Brew.

**Resources:** Sonia Bustamante, Gayathri Sundaram, Bryce Vissel, Bruce J. Brew.

**Supervision:** Bruce J. Brew.

**Validation:** Benjamin S. Summers, Luke Milham, Sonia Bustamante.

**Visualization:** Sonia Bustamante, Krishan Gondal, Michael D. Lovelace.

**Writing – original draft:** Benjamin S. Summers, Luke Milham, Sonia Bustamante, Peggy Rentsch, Gayathri Sundaram, Michael D. Lovelace, Bryce Vissel, Bruce J. Brew.

**Writing – review & editing:** Benjamin S. Summers, Luke Milham, Sonia Bustamante, Krishan Gondal, Peggy Rentsch, Gayathri Sundaram, Michael D. Lovelace, Bryce Vissel, Bruce J. Brew.

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
