## [Decision Letter · Decision Letter 0]

24 Jun 2025

Dear Dr. Brew,

Thank you for submitting your manuscript to PLOS ONE. After careful consideration, we feel that it has merit but does not fully meet PLOS ONE’s publication criteria as it currently stands. Therefore, we invite you to submit a revised version of the manuscript that addresses the points raised during the review process.

We look forward to receiving your revised manuscript.

Kind regards,

Ewa Tomaszewska, DVM Ph.D

Academic Editor

PLOS ONE

Journal Requirements:

4. To comply with PLOS ONE submissions requirements, in your Methods section, please provide additional information regarding the experiments involving animals and ensure you have included details on (1) methods of sacrifice, and (2) efforts to alleviate suffering.

5. We note that you have indicated that there are restrictions to data sharing for this study. PLOS only allows data to be available upon request if there are legal or ethical restrictions on sharing data publicly. For more information on unacceptable data access restrictions, please see http://journals.plos.org/plosone/s/data-availability#loc-unacceptable-data-access-restrictions.

7. Please include a caption for figure 3.

Reviewers' comments:

Reviewer's Responses to Questions

**Comments to the Author**

1. Is the manuscript technically sound, and do the data support the conclusions?

Reviewer #1: Yes

Reviewer #2: Yes

2. Has the statistical analysis been performed appropriately and rigorously?

Reviewer #1: Yes

Reviewer #2: Yes

3. Have the authors made all data underlying the findings in their manuscript fully available?

Reviewer #1: Yes

Reviewer #2: Yes

4. Is the manuscript presented in an intelligible fashion and written in standard English?

Reviewer #1: Yes

Reviewer #2: Yes

Reviewer #1: Dear Authors

I appreciate the thorough research and the clear organization of your ideas. As a reviewer, my goal is to provide constructive feedback to help strengthen your article and enhance its impact. While reading your article, I noticed a few areas that could benefit from further development. You will see a list of comments following. I encourage you to consider these suggestions as you continue refining your work.

Section/subsection: Introduction

Page: 13

Manuscript: 5 weeks following induction of knockout in adult mice, metabolic profiling confirmed significant alterations in the circulating blood concentrations of several KP metabolites, with KMO knockout significantly elevating upstream metabolites KYN, KYNA, and AA. QPRT knockout elevated upstream metabolite QUIN. Interestingly, downstream metabolite levels were unaffected in both models. Despite these metabolic perturbations, histological analysis indicated no overt changes in liver or kidney tissue architecture. Furthermore, comprehensive hematology and blood chemistry panels showed no significant differences in blood cell parameters or organ function versus controls. These findings indicate that in a system free of exogenous inflammatory stimuli, sustained elevations of several KP metabolites do not produce overt renal or hepatic dysfunction.

Comment/Question: Normally, in Introduction section, we try to introduce the research and state the importance of conducting it. We show discrepancies and gaps in previous studies and that we are going to fill the gaps. As readers, we usually don't expect to know about results of the upcoming research. It is recommended to delete this part of introduction.

Section/subsection: Methods

Page: 14

Manuscript: Mice were group housed (2-5 per cage) in individually ventilated cages, …

Comment/Question: In case of control and transgenic mice, although it has been stated in figure caption, it is recommended to state the following issues in methods section as well.

1) How many mice were allocated to each group (Control, KMO, and QPRT)?

2) How many mice were male and how many mice were female?

Section/subsection: Discussion

Page: 36

Manuscript: However, increased blood content of other proteins such as BUN and globulin would be expected following hemoconcentration (62), which we did not observe.

Comment/Question:

1) BUN is not a protein

2) could we think that we had decrease in globulins mainly gamma globulins due to leukopenia induced in transgenic mice while compensated by dehydration and didn't show any change?

Reviewer #2: Some comments and questions are inserted (as STICKY NOTES) throughout the manuscript , to be answered.

- On which bases, the selected Tamoxifen for the experiment, and adjusted the dose

- For Blood collection, processing, and profiling, they didn't mention : For Analytical chemistry, Analyzed immediately or stored until analyzed (Temp. & duration)???

- For Tissue Collection: they didn't mention : From all groups of the mice or representative sample of each group of mice, For each procedure???

**Do you want your identity to be public for this peer review?** For information about this choice, including consent withdrawal, please see our Privacy Policy

Reviewer #1: **Yes: ** Seyed-Hosein Abtahi-Eivary

Reviewer #2: **Yes: ** Prof. Mohamed Abd Ellatif

---

## [Author Response · Author response to Decision Letter 1]

1 Oct 2025

Please see responses in separate document.

---

## [Decision Letter · Decision Letter 1]

20 Oct 2025

Renal and hepatic function is preserved following inducible knockout of kynurenine pathway enzymes KMO or QPRT in adult mice

PONE-D-25-25738R1

Dear Dr. Bruce james Brew,

We’re pleased to inform you that your manuscript has been judged scientifically suitable for publication and will be formally accepted for publication once it meets all outstanding technical requirements.

Kind regards,

Ewa Tomaszewska, DVM Ph.D

Academic Editor

PLOS ONE

Additional Editor Comments (optional):

Reviewers' comments:

Reviewer's Responses to Questions

**Comments to the Author**

Reviewer #1: All comments have been addressed

Reviewer #2: All comments have been addressed

2. Is the manuscript technically sound, and do the data support the conclusions?

Reviewer #1: Yes

Reviewer #2: Yes

3. Has the statistical analysis been performed appropriately and rigorously?

Reviewer #1: Yes

Reviewer #2: I Don't Know

4. Have the authors made all data underlying the findings in their manuscript fully available?

Reviewer #1: Yes

Reviewer #2: Yes

5. Is the manuscript presented in an intelligible fashion and written in standard English?

Reviewer #1: Yes

Reviewer #2: Yes

Reviewer #1: Dear Authors

Thank you for trying to consider nearly all my comments and wish you all success in your carreer.

Reviewer #2: The authors have adequately addressed the comments raised in a previous round of review, and I feel that this manuscript is now acceptable for publication.

**Do you want your identity to be public for this peer review?** For information about this choice, including consent withdrawal, please see our Privacy Policy

Reviewer #1: **Yes: ** Dear Editor,

I want my identity to be public for this peer review.

Reviewer #2: **Yes: ** Prof. Mohamed Abd Ellatif

---

## [Editor Report · Acceptance letter]

PONE-D-25-25738R1

PLOS ONE

Dear Dr. Brew,

I'm pleased to inform you that your manuscript has been deemed suitable for publication in PLOS ONE. Congratulations! Your manuscript is now being handed over to our production team.

Kind regards,

on behalf of

Professor Ewa Tomaszewska

Academic Editor

PLOS ONE